# Mitochondria-Targeted, Nanoparticle-Based Drug-Delivery Systems: Therapeutics for Mitochondrial Disorders

**DOI:** 10.3390/life12050657

**Published:** 2022-04-29

**Authors:** Sakshi Buchke, Muskan Sharma, Anusuiya Bora, Maitrali Relekar, Piyush Bhanu, Jitendra Kumar

**Affiliations:** 1Department of Bioscience and Biotechnology, Banasthali Vidyapith, Vanasthali Road, Dist, Tonk 304022, India; sakshibuchke123@gmail.com (S.B.); muskansharma0041196@gmail.com (M.S.); 2School of BioSciences and Technology, Vellore Institute of Technology (VIT), Vellore Campus, Tiruvalam Road, Katpadi, Vellore 632014, India; bora.anusuiya818@gmail.com; 3KEM Hospital Research Centre, KEM Hospital, Rasta Peth, Pune 411011, India; maitralirelekar@gmail.com; 4Xome Life Sciences, Bangalore Bioinnovation Centre (BBC), Helix Biotech Park, Electronics City Phase 1, Bengaluru 560100, India; pb.inresearch@gmail.com; 5Bangalore Bioinnovation Centre (BBC), Helix Biotech Park, Electronics City Phase 1, Bengaluru 560100, India

**Keywords:** mitochondrial dysfunction, ROS, NP, cancer, Alzheimer disease, diabetes mellitus, ischemia-reperfusion injury

## Abstract

Apart from ATP generation, mitochondria are involved in a wide range of functions, making them one of the most prominent organelles of the human cell. Mitochondrial dysfunction is involved in the pathophysiology of several diseases, such as cancer, neurodegenerative diseases, cardiovascular diseases, and metabolic disorders. This makes it a target for a variety of therapeutics for the diagnosis and treatment of these diseases. The use of nanoparticles to target mitochondria has significant importance in modern times because they provide promising ways to deliver drug payloads to the mitochondria by overcoming challenges, such as low solubility and poor bioavailability, and also resolve the issues of the poor biodistribution of drugs and pharmacokinetics with increased specificity. This review assesses nanoparticle-based drug-delivery systems, such as liposomes, DQAsome, MITO-Porters, micelles, polymeric and metal nanocarriers, as well as quantum dots, as mitochondria-targeted strategies and discusses them as a treatment for mitochondrial disorders.

## 1. Introduction

Mitochondria play a key role in the production of metabolic energy in eukaryotic cells [1]. However, apart from energy production, mitochondria also perform several other functions, namely, calcium signaling, cell proliferation, cell cycle regulation, and apoptosis (Figure 1) [2]. With growing interest in mitochondria, significant efforts are being made in mitochondria-targeting pharmaceutical interventions, resulting in ‘mitochondrial medicine’ as an emerging area of healthcare research. Mitochondria-targeting nanoparticles (NPs) are now a promising field of drug-delivery systems.

Targeting therapeutics to the mitochondria is a challenging task since the mitochondria have four parts, namely, the outer mitochondrial membrane (OMM), the inner mitochondrial membrane (IMM), the intermembrane space (IMS), and the mitochondrial matrix. One of the main hurdles encountered by the molecules in reaching the mitochondrial matrix is the complex membrane network of the mitochondria. Although the therapeutic moieties can pass through the OMM by passive diffusion, phospholipid cardiolipin and a high negative membrane potential cause difficulty for molecules to pass the mitochondrial membranes. The IMM consists of various proteins and ion transporters which play a key role in the electron transport chain and ATP generation [3]. The multiple roles played by the mitochondria make it a target for a variety of therapeutics for the diagnosis and treatment of several diseases.

Low solubility, poor bioavailability, and nonselective biodistribution are some of the present medications’ drawbacks. NPs and conventional chemotherapeutic medicines have recently been combined to produce biocompatible, multifunctional mitochondria-targeted nanoplatforms. Hence, the most promising NP-based techniques for targeting mitochondria in various diseases have been reviewed. This method is now being used to develop targeted medication delivery systems, as well as hybrid nanostructures that can be triggered with light (also known as photodynamic and/or photothermal therapy). The specific delivery of NPs to mitochondria provides an ingenious shortcut to disease treatment that is more selective, precise, and safer. It has the potential to overcome drug resistance while also involving fewer side effects [4].

## 2. Mitochondrial Dysfunction

Mitochondrial dysfunction can be caused by several factors, such as a defect in the electron transport chain and a reduction in the production of ATP [5]. Reactive oxygen species (ROS) produced by the mitochondria cause the majority of the damage [6]. Mitochondrial dysfunction can occur either because of mutations in the mitochondrial proteins or ROS [7]. When mitochondrial proteins are damaged directly, their affinity for substrates or coenzymes is reduced, and as a result, their activity decreases [8]. Consequently, when a mitochondrion is injured, the cellular requirements for energy-repair processes rise, compromising mitochondrial performance even further [9]. Hyperglycemia causes endothelial cells to produce mitochondrial superoxide, which is a key mediator of diabetes and may lead to consequences such as cardiovascular diseases [10]. Heart failure, atherosclerosis, hypertension, ageing, sepsis, ischemia-reperfusion damage, and hypercholesterolemia are all exacerbated by endothelial superoxide generation [11]. Tumor necrosis factor-alpha (TNF-α), a type of inflammatory mediator, has been linked to mitochondrial dysfunction and enhanced ROS formation in vitro [12]. Mitochondrial dysfunction is a result of metabolic imbalance. Vitamins, minerals, and other metabolites are essential cofactors for the synthesis and activity of mitochondrial enzymes. As a result, micronutrient-deficient diets have been linked to mitochondrial degradation and dementia [13]. Mitochondrial dysfunction is the hallmark of several diseases, such as cancer, neurodegenerative disorders, and cardiovascular disorders [14].

Since the first example was published in 1962, that being a 35-year-old female who suffered from the uncoupling of oxidative phosphorylation (OXPHOS), mitochondrial dysfunction has been linked to practically all pathologic and toxicologic disorders [15]. 

Diabetes, huntington disease, cancer, alzheimer’s, parkinson’s, schizophrenia, aging, anxiety disorders, cardiovascular diseases, sarcopenia and exercise intolerance are a few acquired conditions associated with mitochondrial dysfunction while, some of the hereditary conditions associated with it include Kearns-Sayre syndrome (KSS), leber hereditary optic neuropathy (LHON), mitochondrial encephalomyopathy, lactic acidosis, and stroke-like syndrome (MELAS), myoclonic epilepsy and ragged-red fibers (MERRF), Leigh syndrome subacute sclerosing encephalopathy, Neuropathy, ataxia, retinitis pigmentosa, and ptosis (NARP), and Myoneurogenic gastrointestinal encephalopathy (MNGIE) [16,17]. 

### 2.1. Mitochondrial Biogenesis

Mitochondrial biogenesis refers to the process of producing the new mitochondria in cells [18]. Mitochondrial biogenesis depends on mitochondrial, as well as nuclear, factors. For the formation of new organelles, transcription and translation of both mitochondrial and nuclear genes are required [19]. The expression of genes involved in mitochondrial biogenesis is regulated by nuclear transcription factors. These include the nuclear respiratory factors (NRF1 and NRF2), which play key roles in regulating mitochondrial biogenesis. One of the major regulators of mitochondrial biogenesis is a co-transcriptional regulation factor, PGC-1α, which promotes the expression of Tfam by activating NRF1 and NRF2, which in turn aids in the transcription and replication of mitochondrial DNA (mtDNA) [19].

Several posttranscriptional modifications regulate the activity of PGC-1α, including the phosphorylation of PGC-1α by AMPK and the deacetylation of PGC-1α by silent information regulator two (SIR2) protein 1 (Sirtuin 1, SIRT1) [20].

Drugs interfere with the transcription factor pathways, thus affecting mitochondrial biogenesis [21]. Resveratrol (RSV) promotes mitochondrial biogenesis by activating SIRT1, which causes the acetylation and activation of PGC-1α, or it can also activate AMPK, thus activating PGC-1α independently of SIRT1 [22]. AMPK activator 5-aminoimidazole-4-carboxamide ribonucleotide (AICAR) stimulates mitochondrial biogenesis by increasing PGC-1α expression by activating AMPK, and it also mediates mitochondrial apoptosis via the PGC-1α/TFAM pathway in tumor tissues by AMPK phosphorylation [23]. Bezafibrate stimulates mitochondrial biogenesis by activating PPARs and upregulating the PGC-1α coactivator [24].

### 2.2. Mitochondrial Dynamics

Mitochondrial dynamics involve the processes of fission and fusion [25]. Any dysregulation in these processes leads to either a fragmented network of several pieces of mitochondria or a fused network of several, highly connected mitochondria [26]. Mitochondrial dynamics are affected in several diseases, such as neurodegenerative diseases and cancer [27]. There are several proteins playing key roles in fission and fusion. Proteins involved in mitochondrial fission include dynamin-related/-like protein 1 (Drp1), dynamin 2 (Dnm2), and Fis1. Drp1 forms spiral polymers that surround the mitochondrial membrane and initiate fission by GTP hydrolysis [26,28]. Mitochondrial proteins involved in mitochondrial fusion include mitofusin 1 (Mfn 1), mitofusin 2 (Mfn2), and optic atrophy 1 (OPA1) which involves the fusing together of the outer and the inner mitochondrial membranes [26].

Mdivi-1 was the first Drp1 selective inhibitor [29]. Mdivi-1 shows antiapoptotic activities by inhibiting the mitochondrial membrane permeability. It mitigates cell death in tubular and kidney injury by blocking mitochondrial fission [30]. Another selective inhibitor of Drp2 is P110, which inhibits the activation of Drp1 and the interaction of Drp1 with Fis1, which is required for mitochondrial fission in cultured neuronal cells, thus decreasing mitochondrial fragmentation, ROS production, and neurotoxicity [31].

### 2.3. Mitochondria-Related Oxidative Stress

ROS are generated as a byproduct due to the regular metabolism of oxygen involving the superoxide anion (O^2−^), hydrogen peroxide (H_2_O_2_), and hydroxyl radicals (OH^0^) [32,33].

Various autoimmune disorders, cardiovascular disorders, and neurological disorders involve mitochondrial ROS (mtROS) since mitochondria are one of the major sites for the generation of ROS as it activates the RIG-I-like receptors (RLRs), inflammasomes, and mitogen-activated protein kinases (MAPK), which leads to the production of inflammatory cytokines and innate immune responses [34]. MtROS is produced during OXPHOS, and it causes mitochondrial dysfunction by interacting with several components, such as DNA, lipids, and proteins [32].

There are various endogenous antioxidants, such as superoxide dismutase, catalase, and glutathione reductase, which aid in controlling ROS [35]. Coenzyme Q10 (CoQ10)is a lipid antioxidant that aids in preventing the formation of free radicals and any alterations in proteins, lipids, and DNA [36]. Another antioxidant, ferritin, plays a crucial role in sequestering potentially toxic, labile iron and is also involved in ROS-generating Fenton reactions [37,38]. Alpha-lipoic acid is a biological antioxidant that plays a key role in mitochondrial dehydrogenase reactions and recycles vitamin E by interacting with vitamin C and glutathione, thus protecting the membranes [39]. Bilirubin, uric acid, and albumin are other natural antioxidants that scavenge free radicals to prevent oxidative injuries [40,41]. However, it is not easy for the antioxidants to obtain the desired effect because of their restricted distribution in mitochondria. Thus, one of the ways to solve this problem is to attach these antioxidants to triphenylphosphine (TPP+), which can bind with the mitochondrial membrane and result in the severe accumulation of these antioxidants in the mitochondrial matrix [42]. Antioxidants, such as ubiquinone and plastoquinone, have been targeted to the mitochondria in conjugation with TPP+ [43]. Tiron is a mitochondria-localized antioxidant that accumulates within the mitochondria by penetrating the mitochondrial membrane, and it has been found to be effective against UVR-induced oxidative damage [44]. Hemigramicidin-2,2,6,6-tetramethylpiperidine-1-oxyl (Hemigramicidin TEMPO) is a mitochondrion-targeting antioxidant which consists of gramicidin-S and which can be targeted to the mitochondria, independently of the membrane potential and a ROS scavenger TEMPO [45]. Targeting ROS production in mitochondria triggered by SARS-CoV-2 infection may have great potential in drug development against Coronavirus [46].

### 2.4. Mitochondria Induced Apoptosis

Mitochondria plays a very critical role in the activation of apoptosis in mammalian cells, and it is also involved in other functions, such as energy metabolism, calcium homeostasis, and redox regulation. Thus, mitochondria can be potentially targeted in cancer cells using pharmacological agents as a therapeutic approach [47,48]. One effective way to target and diminish cancer cells is to induce apoptosis. Caspase protease, an enzyme, plays a key role in initiating the apoptosis of a cell. Once the enzyme is active, caspases help in cleaving different types of proteins that ultimately lead to rapid cell death [49]. The most common route of initiating caspase activity is through the mitochondrial pathway. This is achieved via the event of mitochondrial outer-membrane permeabilization (MOMP). Once MOMP occurs, mitochondrial intermembrane space proteins, such as cytochrome-c, are released into the cytosol. These proteins further help in the activation of caspase. Apoptosome formation occurs due to the binding of cytochrome-c in the cytosol with an adaptor molecule called APAF-1. Utilization of death-receptor ligands, such as the TNF-related apoptosis-inducing ligand (TRAIL) to initiate the extrinsic apoptotic pathway can also be a route for inducing mitochondrial apoptosis. Apoptosome helps in the activation of caspases enzymes. The MOMP process is highly effective, which leads to ultimate cell death. It can be highly regulated by the Bcl-2 protein family [50].

### 2.5. Mitochondria-Related Cell Signaling

Apart from the diverse roles played by mitochondria, they are also involved in the cell-signaling circuitry. They are involved in cell-signaling in two ways: first, by acting as platforms for protein−protein signaling interactions, and second, by regulating levels of intracellular signaling molecules. Mitochondria can affect major signaling mediators, such as ROS, and thus, they are involved in controlling various signaling processes. Mitochondria have several pathophysiological roles because of their involvement in regulating these processes [51]. Mitochondria are also linked to different innate immune-signaling pathways. The most common is the cytosolic RNA-sensing pathway. In this, mitochondria function as an essential platform on which the reactions can take place. During the process of the apoptosis of cells, oxidized mitochondrial DNA activates the NLRP3 inflammasome to activate inflammatory responses [51,52]. Mitochondria also play an important function in the activation and supply of the energy requirements of functioning T cells and macrophages [52].

## 3. Mitochondria-Targeted, NP-Based Drug Delivery

Mitochondrial dysfunction has been associated with several pathologies that can occur due to alterations in mitochondria-related molecular mechanisms, such as mitochondrial biogenesis, dynamics, mitophagy, and energy metabolism, among other processes.

The steps carried out by the NPs for the transport of the drugs to the mitochondria are shown in Figure 2. The first step involves the intracellular uptake, in which the positively charged NPs bind with the negatively charged phospholipids of the cell membrane. This is followed by endolysosome formation. The endolysosomal membrane then ruptures, causing the release of the contents into the cytoplasm, and the intracellular targeting of the mitochondria takes place [53].

Various barriers within the cell and the mitochondria can be overcome through the design of mitochondria-targeted nanocarriers which have the ability to deliver the drugs selectively to the mitochondria. These nanocarriers aid in the protection of the drug payloads from their elimination and degradation in vivo [7]. Various NP-based drug-delivery systems that can be used for targeting mitochondrial diseases are discussed in this review.

### 3.1. Liposomes

Liposomes are spherical bodies consisting of vesicles made up of phospholipids that contain one or more lipid bilayers and cholesterol. They have an aqueous center that is enclosed within the lipid bilayer [54]. There are around 12 liposome-based drugs available on the market [55]. Liposomes can incorporate both hydrophilic and hydrophobic drugs [54]. Liposomes are used in drug delivery because they are biodegradable, biologically compatible, nontoxic, have the capacity for self-assembly, can carry large drugs, and have several properties which can be altered to control their biological characteristics [7,56].

There are several methods of loading drugs into the liposomes, such as entrapping them in the aqueous region of the liposomes or in the lipophilic bilayers or adsorbing them onto the liposome surface using electrostatic attraction (Figure 3A) [57]. The delivery of therapeutic moieties by mitochondria-targeted, liposome-based drug-delivery systems augments the efficacy of drugs in both in vitro and in vivo models. They are targeted to the mitochondria to encapsulate mitochondria-targeting molecules in lipid bilayers [7].

### 3.2. Liposome-like Vesicles: DQAsomes

DQAsomes are capable of transporting drugs and DNA to mitochondria [3]. These were the first mitochondria-targeted, vesicular, nanocarrier systems [58]. DeQAlinium (1,1′-Decamethylene bis(4-aminoquinaldinium chloride)) (DQA) consists of two quinaldinium rings linked by ten methylene groups which assemble into liposome-like vesicles called DQAsomes (DeQAlinium-based liposomes) [58]. The mitochondrial membranes of the malignant cells possess a negative electrochemical gradient, in response to which the DQAsomes accumulate in the mitochondria [53].

The plasmid DNA can be delivered into the mitochondria via nonspecific endocytic pathways with the help of DQAsomes. The apoptosis or necrosis is achieved by DQAsomes by hampering the mitochondrial transmembrane potential and disrupting the synthesis of ATP, ROS production, and the activation of MAPK pathways, which ultimately leads to the caspase-dependent apoptotic pathway [53]. Weissig et al., showed that a dicationic amphiphilic compound, DQA, forms liposome-like vesicles called DQAsomes. The plasmid DNA pGL3 firefly luciferase was incorporated into the DQAsomes, and these had transfection efficiencies comparable to those of Lipofectin™. The anticarcinoma activity and selective accumulation of the drug DQA in mitochondria make DQAsome a unique drug-delivery system [59].

### 3.3. MITO-Porters

A strategy targeting the mitochondrial genome would be effective in delivering the therapeutic agents to the mitochondrial matrix containing the mtDNA pool. MITO-Porter is a liposome-based carrier that delivers macromolecules efficiently to the cytoplasm [60,61], as well as to mitochondria, by membrane fusion [62].

In 2008, Yamada et al. decided to deliver green fluorescence protein (GFP) to rat-liver mitochondria via the membrane fusion mechanism to overcome the limitations of the method by which mitochondrial targeting signal (MTS) peptide is conjugated with exogenous proteins and small linear DNA in order to facilitate their delivery to mitochondria. The plasma membrane is the initial line of defense against intracellular targeting. Yamada et al. coated the MITO-Porter surface with high-density octaarginine (R8), resulting in macropinocytosis instead of clathrin-mediated endocytosis, which allowed particles to enter the cell without being damaged. The MITO-Porter attaches to the mitochondrial membrane via electrostatic interactions after being released from macropinosomes, causing the MITO-Porter and mitochondrion to fuse. The R8-liposomes’ lipid content is crucial since it has to fuse with the mitochondrial membrane. Therefore, the MITO-Porter system is based on the identification of two extremely fusogenic lipid compositions: sphingomyelin (SM) and phosphatidic acid (PA) [62].

In a study by Yasuzaki and colleagues, the aim was to deliver the MITO-Porter carrying the moiety to the mitochondrial matrix in rat-liver mitochondria. Here, membrane-impermeable, red-fluorescent propidium iodide dye used to stain nucleic acids was incorporated into the MITO-Porter liposomes and delivered to the mitochondrial matrix. It was found that the reason behind the localization of these liposomes within the mitochondria was the electrostatic interactions, and the fusogenic lipid components were responsible for the fusion of these liposomes with the mitochondrial membranes [60]. This approach is being researched further for mitochondrial gene therapy [63] and photodynamic cancer therapy [64].

### 3.4. Micelles

Micelles are amphiphilic molecular systems that are spherical in shape, containing a hydrophobic head group and a hydrophilic tail (Figure 3C). Micelles typically range from 10 to 100 nm in size. These particular types of molecules are formed in aqueous solutions whereby the polar region that is hydrophilic faces the external surface, and the nonpolar region faces the interior part [65,66]. Micelles have garnered attention due to their ability to encapsulate drug substances that are less soluble in water or aqueous environments. Such molecular arrangements are also known as ‘polymeric micelles’, which can deliver drugs of low water-solubility. Micelles have been utilized for mitochondrial-targeted cancer therapies. This drug-delivery mechanism also aids in improving the bioavailability of the drug. The bioavailability of micelles containing drugs can be enhanced by adding external surfactants [66].

### 3.5. Polymeric NPs

Polymeric NPs are biodegradable and biocompatible and can be targeted to mitochondria for drug delivery because they are easy to manufacture; thus, surface modifications can be made easily and are tunable to drug-release profiles [7]. Polymeric NPs have the properties of higher stability and a more controlled payload release, which are not found in liposomes [3].

Various polymers, such as poly(lactic acid) (PLA), poly(glycolic acid) (PGA), poly(lactic-co-glycolic acid) (PLGA), and polycaprolactone(PCL), can be used for drug delivery and can be turned into NPs that can incorporate both water-soluble and -insoluble payloads via engineering polyethylene glycol(PEG) to hydrophobic blocks with the help of processes such as emulsification−solvent evaporation or nanoprecipitation. Hydrophobic blocks aid in augmenting stability, whereas PEG helps in increasing the residence time in vivo [7,67].

### 3.6. Dendrimers

Dendrimers are macromolecules consisting of a central core and multiple branches. These branches have various ligands attached to them on their peripheries. With increased branching, generation numbers, such as G1, G2, G3, G4, and so on, continue to be associated with them (Figure 3B). There are various types of dendrimers, such as peptide dendrimers, liquid crystalline dendrimers, tecto dendrimers, chiral dendrimers, glycodendrimers, polyamidoamine (PAMAM) dendrimers, PAMAM organosilicon dendrimers (PAMAMOS), etc. [68]. Dendrimers are potential nanocarriers for various therapeutic drugs. They are known to show various pharmacokinetic properties, one of which is transdermal drug delivery. It overcomes the side effects of orally given NSAIDs. Several projects are being conducted to use dendrimers in gene therapy as well, the main aim of which is to use dendrimers as potent gene-delivery systems to the cell without causing any harm to the DNA [69]. Dendrimers, when conjugated with lipophilic cations, such as rhodamine or TPP, can deliver drugs to mitochondria via exhibiting endosomal escape properties [3].

Biswas et al., designed the mitochondria-targeted PAMAM dendrimer (G(5)-D). It was prepared in conjunction with TPP on its surface. It was found that these NPs were taken up by the cells efficiently and showed good mitochondria-targeting activity [70]. PAMAM dendrimers, along with TPP, are widely used in targeting mitochondria for drug delivery. In addition, these dendrimers have been proven non-toxic during the process of transfection [3].

### 3.7. Metal NPs

Metal NPs, such as silver and gold, possess properties such as SPR (surface plasmon resonance) which is not found in liposomes, dendrimers, and micelles. They are also biocompatible and versatile in terms of surface functionalization. Gold NPs (AuNPs) can be conjugated to drugs by ionic or covalent bonds and physical absorption and act as a drug-delivery system, whereas studies have also used silver NPs for the release of ornidazole, which has an in vitro release of 98.5% [71]. Furthermore, gold, silver, and titanium dioxide have been used significantly in recent years as therapeutic tools, and they can be designed to produce very small-sized (3–30 nm), homogenous NPs [7]. Metal NPs are used as potential therapeutics for various infectious diseases because, apart from having good physicochemical properties and surface charges, they also have the ability to conjugate with drugs, antibodies, and proteins, which provides them protection against the host’s immune system, thus increasing their circulation time in the blood [72].

AuNPs have been used as core components while preparing NPs because of their bioinertness, ease of synthesis, and characterization [73]. AuNPs, along with turbo-green fluorescent protein (TGFP) conjugate, have shown an apoptotic effect on breast cancer cells by partially rupturing their mitochondria, and hence, such NPs are helpful in photochemical therapy for cancer [74].

Iron NPs have been used to trigger autophagy in cancerous cells by targeting their mitochondrial DNA [75]. Silver NPs (AgNPs) have also shown potential as nanocarriers for the treatment of various diseases. They have unique antimicrobial, antiviral, and antibacterial properties. Along with this, AgNPs, with or without conjugates, have been seen as potent drug carriers for treating cancer because of their antitumor properties [76].

### 3.8. Quantum Dots

Quantum dots (QDs) are also known as ‘nanoscale semiconductor crystals’, and they display luminescence. They possess a metalloid crystalline core. It can be used in fluorescence technology, which depends on factors, such as composition and size. QDs in general can range from nanometers to microns. For better fluorescence, the core of QDs can be composed using different materials, such as cadmium–selenium (CdSe), indium–phosphate (InP), etc [77]. QDs can be manufactured using a nanofabrication technology in which properties, such as size, shape, and molecular interactions, can be defined as per requirements [78]. Since QDs are mostly categorized as NPs, they can be advantageous as efficient drug-delivery molecules. Due to their small size, QDs have a larger surface area which can be modified according to biological conditions to lower the aggressive immune response. Moreover, QDs also provide good pharmacokinetic properties [79]. QDs can be coated with biocompatible materials and polymeric materials to enhance solubility and bioactivity once present in vivo. Materials, such as polyhedral oligomeric silsesquioxane-polycarbonateurethane (POSS-PCU) [80] and PEG, can be used [79]. However, metalloid cores can be toxic for human usage. They can be lethal or carcinogenic [81]. Researching biocompatible materials for quantum dots is the current need.

QDs can be utilized in targeted delivery applications. They are usually conjugated with ligands that can be recognized by biological immune systems, such as antibodies, DNA, biotin, streptavidin, and peptides [82]. There are two methods for utilizing QDs as drug-delivery vehicles. First, the drug molecule can be attached to the surface of the QDs. The drugs can be transported to a specific site and can be released when they experience biological phenomena, such as the presence of enzymes. Second, drugs can be encapsulated in a QD−NP system. The entire NP is then transported for site-specific actions. They can be subsequently exposed to conditions where the NP’s outer covering may become degraded or diffused [83]. This was first demonstrated by Bagalkot et al. when they formulated the “QD−Apt(DOX)” complex (QD−aptamer(Apt)−doxorubicin (DOX) as a targeted treatment material for cancer imaging, therapy, and sensing using doxorubicin (DOX),which is an anticancer drug [84].

Carbon QDs or carbon dots are NPs that are fluorescent in nature. They possess efficient optical properties, are photostable in nature, and are highly biocompatible in nature. Due to these attributes, they are usually used in different processes, such as bioimaging, biosensing, and cancer therapy. They also have various functional groups on their cell surfaces which enable the modification of specific functional groups to enhance the process of target-specific drug delivery. Carbon dots, when conjugated with Rose Bengal (RB), a common photosensitizer, form a complex that aids in the cellular uptake of the carbon dot for efficient, mitochondrial-based, photodynamic therapy (PDT) (Figure 3D). Hua et al. demonstrated the production of an efficient carbon dot that can be used for mitochondrial-targeted therapy. They prepared carbon dots via the hydrothermal treatment of a mixture of chitosan, ethylenediamine, and mercaptosuccinic acid. It was also observed and indicated that the cellular uptake of carbon dots is a temperature-dependent process. The temperature at which cells are cultured with carbon dots is an essential factor [85].

**Figure 3 life-12-00657-f003:**
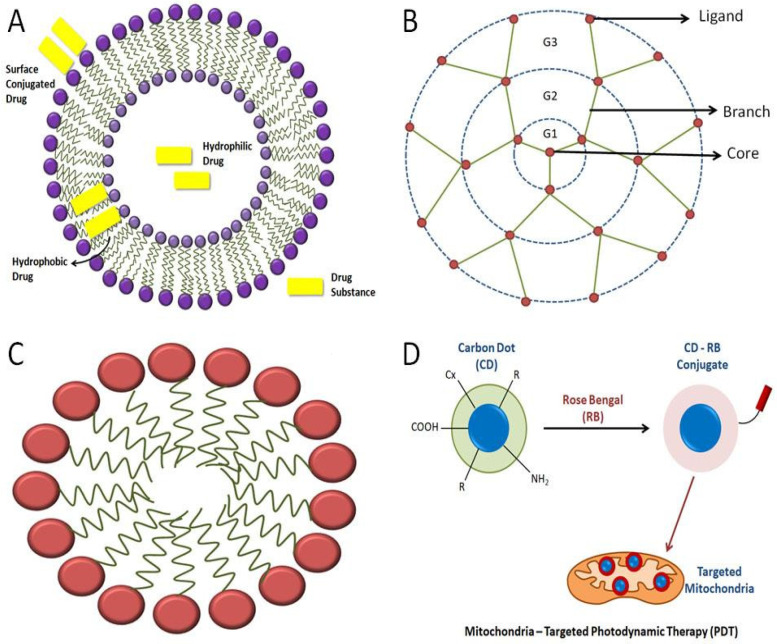
**Representation of various kinds of mitochondria-targeting NPs:** (**A**)various strategies of loading drugs into the liposome; (**B**) basic structure of a dendrimer; (**C**) general structure of a micelle; (**D**) carbon quantum dots used in mitochondrial-targeted therapy using Rose Bengal to bring changes in the conjugated carbon-dot molecule, where “R” is a modifiable functional group on carbon quantum dots’ surface (Adapted from Ref. [85], Hua et al., 2017).

## 4. Targeting Mitochondria for Treating Mitochondrial-Dysfunction-Related Disorders

A few disorders that are associated with mitochondrial dysfunction, such as Alzheimer disease, cancer, diabetes, and ischemia-reperfusion injury are discussed in this review. Mitochondria can thus act as a therapeutic target for the diagnosis and treatment of these diseases, and mitochondrial-targeted drug−NP conjugates can serve as potential therapeutics for these pathologies (Table 1). There are several other pathologies, such as inflammation, protozoal infection, muscular dystrophies, sarcopenia, lung diseases, and colitis, in which mitochondria can also be a therapeutic target [21].

### 4.1. Cancer

Cancer is the second-greatest cause of mortality after cardiovascular diseases and causes around one in six deaths globally (https://www.who.int/news-room/fact-sheets/detail/cancer (accessed on 28 January 2022) and https://www.cancer.gov/about-cancer/understanding/what-is-cancer (accessed on 28 January 2022)). Cancer cells reprogram their metabolisms to meet cell-proliferation and migration demands [108]. Mitochondrial metabolism and mtROS generation are essential for tumorigenesis [109].The relation of cancer with mitochondria was first explained by Otto Heinrich Warburg in the 1920s, and according to the Warburg Effect, the metabolism of cancer cells is reprogrammed from OXPHOS to oxidative glycolysis as there are certain defects in the mitochondrial OXPHOS of cancer cells, and thus they depend on aerobic glycolysis as the major ATP source for cellular proliferation. This is in contrast with the normal cells, which primarily depend on OXPHOS for energy generation [110,111].

Cancer-cell mitochondria have various factors differentiating them from normal-cell mitochondria, which include ATP production, metabolism, mitochondrial membrane potential, ROS production, glutathione level, pH, and oxygen consumption (Figure 4) [112].Cancer cells increase ATP production with the help of three enzymes, namely, alpha-ketoglutarate dehydrogenase (α-KGDH), isocitrate dehydrogenase, and pyruvate dehydrogenase, in which Ca^2+^ is a cofactor. Thus, an increase in mitochondrial Ca^2+^ leads to a simultaneous increase in enzymatic activity, hence increasing ATP production. On the other hand, the increase in mtROS causes genomic instability and metastatic potential, which lead to tumor progression [113].

Cancer cells also have a higher proton-motive force (Δp) than normal cells. The increased production of ATP by glycolysis decreases the Δp utilization for ATP synthesis by OXPHOS. This increased mitochondrial Δp exhibits an increased membrane potential (Δψ) in cancer cells, thus enhancing the uptake of drugs into the mitochondria of cancer cells [21]. In addition, the depletion of ATP, the damaging of mtDNA, and the inhibition of the mitochondrial respiratory pathway can lead to cancer-cell death. A cancer cell is dependent on its mitochondria for ATP production via bioenergetic pathways and macromolecule production via biosynthetic pathways. Similarly, ROS production by mitochondria (redox capacity) is also important for cancer-cell proliferation. Thus, inhibition of these pathways makes mitochondria a major therapeutic target in treating cancer [114,115].

Chemotherapy plays a major role in keeping a check on cancer, but there are also a few obstacles that are faced during chemotherapy, such as multidrug resistance and cancer metastasis caused by this resistance. Over the years, significant research has been performed on mitochondria-targeted drug-delivery nanosystems in the field of cancer research (Figure 5). In a study performed in 2011, topotecan-loaded liposomes were used to target mitochondria. The study was performed on drug-resistant breast cancer cells and metastatic melanoma in the lungs of nude mice. It was observed that mito-targeting, topotecan-loaded liposomes showed a great inhibitory effect on resistant tumor cells, as well as having an antimetastatic effect on the metastatic melanoma model. The results were deduced based on various factors, such as colocalization, enhancement in drug content, the membrane potential of mitochondria, membrane permeability, activation of caspase-9 or -3 leading to the apoptosis of cancerous cells, etc. This study suggested that the use of topotecan-loaded liposomes can prove to be a better cancer-treating strategy by overcoming both drug resistance and related metastasis [86].

A similar approach to treating cancer-drug resistance was used by Wang et al., where a natural plant product PTX (paclitaxel) was used along with NPs to treat cancer by targeting the mitochondria of drug-resistant lung cancer cells. Here, a pharmaceutical adjuvant PF127 was used with TPP NPs, which allowed easy internalization into tumor cells, and, in order to neutralize the cationic property of TPP, anionic hyaluronic acid (HA) was also added. Together, they formed TPP-PF127-HA nanomicelles, also called ‘TPH’. This TPH NP was then loaded with PTX, forming a TPH/PTXA complex that targeted the drug-resistant lung cancer cell mitochondria and induced apoptosis [87].

The use of biological, green synthesis of NPs for cancer treatment is also a current topic of interest, as chemical synthesis can lead to the accumulation of toxic substances on the surface of NPs. George et al. conducted a study using AgNPs and conjugated them with an extract from *Rubus fairholmianus*. These conjugated RAgNpPs were seen to target the mitochondria of breast cancer cells MCF-7 and induced cell death via apoptosis, hence proving that green NPs have great potential in anticancer-drug development [88]. Apart from AgNPs, iron oxide NPs have also been seen to show great anticancer efficacy (Table 2).

The use of dendrimers is also popular in cancer treatment. Kianamiri et al. constructed a generation 4 PAMAM dendrimers with a conjugation of TPP. This dendrimer was used to deliver curcumin, which has excellent anticancer activity. The TPP-PAMAM-curcumin conjugate, when administered to Hepa1–6-tumor-bearing mice, targeted mitochondria and induced apoptosis in cancer cells. In addition, the targeted dendrimer−curcumin, when compared to free curcumin, showed better results in cancer treatment. This proves that dendrimers are efficient nanocarriers for targeting mitochondria in cancer therapy [89]. Various other PAMAM dendrimers are used in cancer treatment (Table 3). Interestingly, the effect of curcumin on mitochondria is dose-dependent. Curcumin concentrations can either protect the mitochondria and increase cell viability or damage it and cause cell death. Curcumin at both low and high concentrations increases the level of ROS in cells. At low concentrations, a very small increase in ROS is observed, which is balanced by the cell antioxidant defense system. The ROS produced here are used to activate the cell redox signaling pathways that further activate the transcription factors involved in mitochondrial biogenesis. At a high concentration of curcumin, the level of ROS is increased drastically, and this increase cannot be balanced by the cell. This leads to oxidative stress and mitochondrial dysfunction by opening the mitochondrial permeability transition pores and releasing cytochrome C, and hence, apoptotic cell death occurs [125].

Yamada Y. et al. performed a study on OS-RC-2 cells, which are DOX renal cancer cells. This study included the use of a MITO-Porter to deliver DOX by targeting the mitochondria of cancer cells. Cell toxicity was evaluated between naked DOX, liposomal DOX, and the DOX−MITO-Porter. It was observed that the DOX in the MITO-Porter was more effective in decreasing cell viability than the naked or liposomal DOX. The reason behind this observation was the specificity of the DOX−MITO-Porter in targeting the mitochondria for DOX delivery and causing mitochondrial toxicity, which leads to the death of drug-resistant renal cancer cells (Figure 6) [90].

Nitric oxide (NO) is also used in anticancer therapy because a high dose of NO inhibits mitochondrial respiration, therefore making it a potential therapeutic target for cancer treatment. Guo et al. used graphene quantum dots in conjugation with TPP and ruthenium nitrosyl as a nanoplatform to deliver NO to the cancer cell mitochondria. This caused a photothermal effect, leading to both in vivo and in vitro anticancer efficacy [91].

Another preoperative strategy for cancer stem-cell eradication by targeting mitochondria was suggested in a clinical pilot study performed in 2018. According to this study, doxycycline, a mitochondria-targeting drug given to breast cancer patients in vivo before surgeries, lead to a decrease in CD4 and ALD1 expression, and hence a clear reduction in cancer stem cells was observed. Further study in this area may lead to a breakthrough in cancer treatment [131].

### 4.2. Alzheimer Disease

Alzheimer disease is one of the most common neurodegenerative diseases. Symptoms of people suffering from Alzheimer disease include memory loss, difficulty in solving problems, decreased judgment, and mood and personality changes. The severity of the disease varies from person to person. Plaques are formed in the brain because of the deposition of beta-amyloid protein, which is then followed by neurofibrillary tangles, which are composed of phosphorylated Tau proteins, the loss of cells, vascular damage, and dementia [132,133]. Treatment of this disease includes the usage of drugs, such as cholinesterase inhibitors (donepezil, rivastigmine, and galantamine) and the N-methyl-D-aspartate-receptor-antagonist memantine [134].

Mitochondrial dysfunction plays a key role in Alzheimer disease. The mitochondrial hypothesis was given by Swerdlow and Khan, which represented the primary pathogenesis in sporadic AD [135]. According to the mitochondrial cascade hypothesis, each individual inherits a certain baseline mitochondrial function, and the mitochondrial function of each individual declines with time. It also mentions that the mitochondrial decline rate is determined by genetic and environmental factors. When the decline in the mitochondrial rate exceeds a certain threshold, then changes related to the histology of Alzheimer disease, such as Aβ deposition, occur. This suggests that individuals whose baseline function and rate of decline of mitochondria are both low or both high will develop the changes related to Alzheimer disease at intermediate ages, while those individuals whose baseline function is low and rate of decline of mitochondria is high will develop these changes at younger ages, and individuals who have a high baseline function and a low rate of decline of mitochondria will develop these changes at comparatively later ages [136]. Thus, mitochondria can act as a potential target for the treatment of Alzheimer disease.

Overproduction of ROS, disruption of Ca^2+^ homeostasis, alteration in mitochondrial dynamics, and ATP generation, as well as mitophagy, can lead to mitochondrial dysfunction in AD. Aβ-induced mitochondrial dysfunction can also be caused due to increased ROS production, which leads to the disruption of the lysosome membrane, causing neuronal death [137]. The drugs and antioxidants have to cross the blood−brain barrier (BBB) as well as the mitochondrial membranes to work efficiently. Thus, NP-based systems have been developed to overcome the BBB and improve efficacy [138].

Solid lipid NPs (SLNs) consist of a lipid core, whereas nanostructured lipid carriers (NLCs) are prepared from a combination of solid and liquid lipids. These lipid carriers can easily penetrate the BBB [139]. Curcumin has therapeutic properties due to its antioxidant and anti-inflammatory nature. It prevents the formation of Aβ plaque by preventing the aggregation of amyloid β peptides. It has also been shown to mitigate neurodegeneration. It is incorporated in NLCs for targeted delivery [93]. Ferulic acid (FA) also has antioxidant and anti-inflammatory properties. It is found in plant cell walls. The administration of FA with SLNs decreases ROS and prevents mitochondrial dysfunction. Thus, it has been suggested as a potential therapy for AD [92].

Nanomicelle preparation of water-soluble CoQ10 (WS-CoQ10) has been shown to decrease mitochondrial dysfunction and ROS levels and produces the resumption of autophagy in the fibroblasts of AD patients [95]. Osthole (Ost), which is a coumarin derivative, is used for the preparation of transferrin-modified Ost liposomes (Tf-Ost-Lip). These liposomes decrease Aβ plaque deposition and oxidative stress, as well as neuroinflammation, in mice with AD [94].

In a study carried out by Marrache et al., NPs were designed by using polymer-blending technology in an attempt to improve their size and surface charge, as not all NPs are capable of crossing the double membrane layer of the mitochondria. PLGA-b-PEG-TPP copolymer was synthesized, and curcumin was used as the cargo. It was demonstrated that these showed augmented protective effects against β-amyloid peptides in human neuroblastoma cells compared to nontargeted particles or free curcumin [96]. NAC (N-acetyl cysteine) is an antioxidant and anti-inflammatory agent. It decreases oxidative stress, thus preventing mitochondrial dysfunction. PAMAM dendrimers can target the activated Mi/Ma and astrocytes at the site of brain injury by crossing the BBB. Targeting NAC to mitochondria by conjugating them to dendrimers has shown neuroprotective effects and thus can be a potential strategy for treating mitochondrial dysfunction related to neurological disorders [97].

Metal NPs are used in the treatment of diseases, such as Alzheimer and Parkinson diseases. They can be used for drug delivery for the treatment of diseases associated with the central nervous system. In these diseases, they are used for the detection of Aβ and α-synuclein whereas, in terms of the treatment, they are used to manipulate the aggregation of Aβ, thus decreasing toxicity [140]. Ceria (CeO_2_) NPs can change reversibly between Ce3+ and Ce4+ oxidation states. 5XFAD transgenic AD mice were studied to determine the effect of TPP-ceria NPs in vivo whereas the in vitro studies were carried out with human neuroblastoma SH-SY5Y cell lines. These NPs were found to be biocompatible and decreased oxidative stress, both in vitro and in vivo, and also suppressed neuronal death [98].

### 4.3. Diabetes Mellitus

Diabetes is a global public health problem and resulted in approximately 2.2 million deaths in 2016 alone. It is a result of the insufficient production of insulin by the pancreas or the inefficiency of the body to utilize produced insulin, resulting in either type1 diabetes or type 2 diabetes, respectively (https://www.who.int/news-room/fact-sheets/detail/diabetes (accessed on 28 January 2022)).

Mitochondrial dysfunction, namely, mitochondrial DNA mutations, decreased ATP production and mitochondrial numbers in skeletal muscle as well as reducing mitochondrial stimulus−secretion coupling in pancreatic beta cells, has been linked to the pathophysiology of chronic metabolic illnesses, such as type 2 diabetes mellitus [141]. It has been suggested in several reports that mitochondrial dysfunction is related to insulin resistance and β-cell dysfunction [142]. The production of ROS by mitochondria and other cellular sources may disrupt insulin signaling in muscle, resulting in insulin resistance [143]. The accumulation of intramuscular fat, insulin resistance, and muscle dysfunction in older people is caused by a reduction in mitochondrial oxidative capability combined with an increase in ROS formation [144]. Increased acetyl coenzyme A (CoA) and the oxidation of lipids in the mitochondrial citrate inhibit some enzymes using glucose. It thereby reduces the consumption of glucose, which leads to increased intracellular glucose levels, as well as to the decreased production of insulin [67].

When insulin binds to its receptor, it induces glucose uptake. Once the insulin receptor (IR) is activated, it phosphorylates insulin receptor substrate (IRS1), which further leads to the activation of phosphoinositide 3-kinase (PI3K), which in turn activates Akt, which facilitates glucose uptake. Due to mitochondrial dysfunction, excessive production of ROS occurs, which activates serine/threonine kinases, which cause the increased serine phosphorylation of IRS1. This blocks the PI3K activity, further leading to the inhibition of glucose uptake (Figure 7) [145,146].

Mitochondria appear to be a promising target for the development of innovative medicines with the ability to treat insulin resistance at several levels and in various tissues in a more integrated treatment strategy. Multiple medications are used to treat diabetes with pharmacological treatments that target mitochondria, namely, metformin [147,148], thiazolidinediones (TZDs) or glitazones [149], sodium-glucose co-transporter-2 (SGLT-2) inhibitors or gliflozins [150], insulin [151,152], statins [153], simvastatin [154], incretin [155], and so on.

In April 2021, Karunanidhi et al. prepared SNPs, loaded with *Ficus religiosa* L. extract, which were activated by TPP. They administered these particles to oxidative stress-induced diabetic rats orally. The findings revealed an improvement in mitochondrial function in terms of morphology, Ca^2+^ concentration, membrane potential, antioxidants, and Complex I, II, IV, and V activity. Furthermore, reductions in apoptotic markers, blood glucose, and glycosylated hemoglobin were observed, along with improved plasma insulin [99].

According to a study by Tang et al., yttrium oxide NPs (Y_2_O_3_ NPs) have a considerable influence on the decrease of oxidative damage. Y_2_O_3_ NPs changed oxidative-stress-related biochemical markers in many disease models, including diabetes. Although the antioxidant and anti-inflammatory capabilities of Y_2_O_3_ NPs make them a promising antidiabetic drug, additional research is needed to fully understand the pharmacological and toxicological characteristics of these NPs [100].

Ward et al. compared the combined, as well as the individual, effects of ramipril (angiotensin-converting enzyme) and MitoQ on mice with diabetic kidney disease. MitoQ is a modified version of coenzyme Q, and with a lipophilic cation, it specifically enhances its uptake into mitochondria, where it is hypothesized to function as an antioxidant. It enhances renal function by lowering glomerular hyperfiltration, albuminuria, and tubulointerstitial disease. It also reduces elevated levels of ATP, ADP, and the ATP:AMP ratio caused by diabetes. It works as an uncoupler, increasing mitochondrial oxygen demand, while reducing ATP generation. It inhibits the glycerol/fatty acid synthesis precursor DHAP and the glycogenolysis breakdown product glucose-1-phosphate, both of which are induced by diabetes. As a first-line therapy for diabetic kidney disease, MitoQ provides renoprotection equivalent to that of ramipril; however, their combination does not provide synergistic benefits [101].

Chinnaiyan et al. produced metformin-loaded pectin NPs (PCMNPs) using the ionic gelation process. They studied the in vitro hemocompatibility, protein binding stability, and impact on glucose uptake. Because the PCMNPs caused negligible damage to the erythrocyte membrane, there was no significant hemolysis. Furthermore, the positively charged PCMNPs absorbed the negatively charged, free hemoglobins with ease. Glucose tolerance is impaired in diabetic people. RBC glucose uptake was enhanced with these NPs. This was probably due to pectin’s antioxidant capabilities, which reduced lipid peroxidation and oxidative damage in RBCs. Metformin controls blood sugar levels by increasing cellular glucose absorption. Compared to metformin-treated RBCs, PCMNP-treated RBCs increased glucose absorption. During the glucose absorption assay, no erythrocyte membrane deformation was detected at any of the doses evaluated. As a result of the use of this nanoformulation, the bioavailability of metformin can be enhanced, allowing for the more effective treatment of type 2 diabetes [102].

### 4.4. Ischemia-Reperfusion Injury

Ischemia refers to the condition of deficiency of blood supply to tissues of certain regions, most commonly observed in cardiomyocytes. Reperfusion refers to the process of blood supply to ischemic tissues. Even though reperfusion is essential, it has been characterized as a causal event for damage to cardiomyocytes and ultimately causes necrosis. These types of damages are termed ‘ischemia-reperfusion injuries’(IRIs) [156,157]. Ischemia-reperfusion injuries may occur in conditions such as atherosclerosis and acute myocardial infarction. Patients who survive heart attacks may still suffer from ischemia-reperfusion injury and experience damage to cardiomyocytes [157].

The damage caused in cells due to ischemia varies with the extent and duration of the blood-supply deprivation. During ischemia, anaerobic metabolism is dominant. This results in a decrease in pH in the cells. To equilibrate acidity, the Na^+^/H^+^ antiporter excretes excess hydrogen ions, which induces an influx of sodium ions. The depletion of oxygen levels also results in a decrease in cellular ATP. The ATPases become inactive, which ultimately reduces Ca^2+^ efflux out of the cell. This results in the accumulation of Ca^2+^ ions. These changes in ion levels trigger the opening of the mitochondrial permeability transition pore (mPTP). This opening releases the mitochondrial membrane potential, hence impairing the process of ATP production. Ultimately, cellular and ionic changes in the cell environment lead to the activation of intracellular proteases, such as calpains, which damage the myofibril of the cardiomyocytes and cause cell death [157]. These structural changes in the mPTP cause increase in oxygen requirements, and this often leads to the initiation of the autophagy of mitochondria. Once the integral component of the cell disintegrates, the cell is no longer viable (Figure 8) [158]. mPTP also induces an excessive production of ROS [158].

The NP-based treatment of IRI is an essential field that is currently being explored by researchers. Due to the excessive production of ROS, antioxidant peptides entrapped in NPs are an efficient method of treating and preventing reperfusion-induced injuries. RSV, Szeto–Schiller-31 (commonly known as SS-31) and Bendavia/Elamipretide (MTP-131) antioxidant peptides have shown successful uptake by the mitochondrial membrane and hence are being used in mitochondria-targeted therapy [159].

In a study conducted by Cheng et al., a mitochondria-targeted NP was developed and investigated. They developed multistage, continuous, targeted-drug-delivery-carrier NPs (MCTD-NPs) which were used to deliver RSV. The RSV molecule is hydrophobic in nature; hence, it is targeted toward ischemic cells using a ligand, such as IMTP, which helps in binding with ischemic cardiomyocytes. RSV, when released in the mitochondria, was found to induce a therapeutic effect for myocardial ischemia-reperfusion injury. The experimental results indicated that MCTD-NPs significantly helped in inhibiting mPTP opening. A further decrease in the infarct size was also found in in vivo experiments [103].

In a recent research study by Ikeda et al., an NP composition of PLGA was utilized to contain the drug cyclosporine A (CsA), or pitavastatin. It was indicated that the intravenous administration of NPs containing these drugs protected the heart against myocardial IRI by inhibiting mPTP opening and monocyte-mediated inflammation [104].

Apart from being a free radical scavenger, melatonin has both antioxidant and anti-apoptotic properties. Melatonin encapsulated inside PLGA NPs when administered in rat brains with cerebral IRI showed higher potential in comparison to free melatonin in restoring mitochondrial functions. Oxidative stress decreased, and the restoration of restored catalase, superoxide dismutase, glutathione reductase activities, and lipid peroxidation occurred [106]. Ishikita et al. incorporated Mdivi1 in PLGA NPs, which provided cardioprotection in myocardial IRI. They showed that Mdivi1 inhibited MOMP, which is responsible for mitochondria-mediated apoptosis [107]. Quercetin is a dietary flavonoid that has the capability to capture ROS and inhibit several cardiomyopathy-related signaling pathways. In an experiment conducted by Lozano et al., the nanoencapsulation of quercetin in PLGA NPs was achieved. It was observed that the release of quercetin results in a decrease of free oxygen radicals that are induced by antimycin A. Quercetin is successful in showing the improved cell viability of IRI-affected cardiac cells [105].

## 5. Conclusions and Future Perspective

Mitochondria involve a variety of functions, such as calcium signaling, cell growth and differentiation, cell cycle regulation, and apoptosis, apart from energy production. Mitochondrial dysfunction is associated with several neurodegenerative, cardiovascular, and autoimmune diseases, thus making it a potential target for diagnostic and therapeutic interventions for such diseases. Mitochondria-targeted drug−NP conjugates can thus act as potential therapeutics for these diseases.

The therapeutic moieties cannot reach the mitochondria directly as they need to overcome barriers, such as the plasma membrane and the mitochondrial membranes. The properties of NPs to target specific cells and localize within mitochondria aid in the treatment of mitochondrial-dysfunction-related disorders. These drugs overcome the issues of the poor biodistribution of drugs and pharmacokinetics and have increased specificity. Several drug payloads have the ability to target mitochondria; however, for the delivery of these payloads, they need to overcome several cellular and mitochondrial barriers, and many of them are not able to overcome these without any assistance. NPs provide promising ways for targeting drug cargos to mitochondria. NPs based on liposomes, liposome-like vesicles, biodegradable polymeric particles, and metal particles help in delivering the drug payloads to the mitochondria. Although these nanopreparations are successful in in vitro studies, preclinical and clinical studies need to be carried out so as to understand their clinical potential. Still, several studies need to be conducted to understand the safety of these drug-delivery systems in vivo. Enormous efforts are being made in the preparation of mitochondrial-targeted nanomedicine globally; however, a better understanding of the subject is still required so as to use them as potential therapeutics for a wide range of diseases.

The promising results of mitochondrial-targeted drug delivery in the successful treatment of several diseases have also paved the way for drug-free approaches. Recent approaches have shown that anticancer, cytotoxic drugs interfere with DNA and damage it to kill the cells. This has led to the development of a “drug-free” approach, wherein certain macromolecular systems are introduced which cause physical or metabolic damage and cellular apoptosis. The self-assembly of biomolecules, aggregation, and biomineralization will lead to a decrease in ATP and cell survival, as well as the loss of membrane integrity, thereby killing cancer cells. This will facilitate better treatment strategies and overcome drug resistance, thereby targeting cancer cells more effectively. Thus, mitochondria-targeted, drug-free strategies appear to be the future in mitochondrial medicine and therapeutic regeneration modalities for several diseases.

## Figures and Tables

**Figure 1 life-12-00657-f001:**
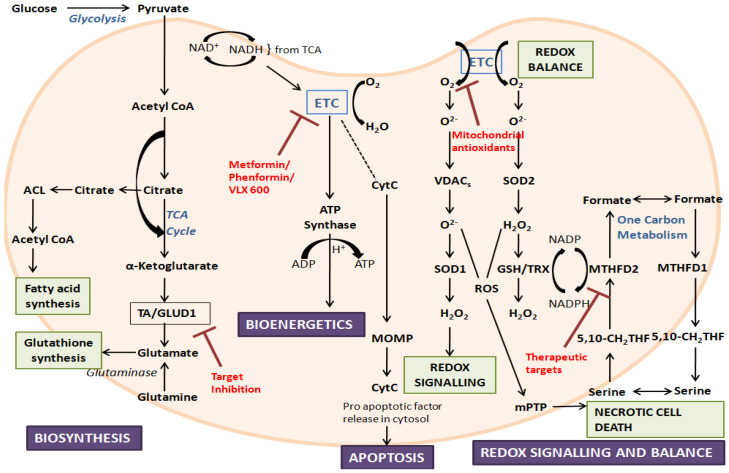
**An overview of mitochondrial functions and their dysfunction, along with some target inhibitors:** The figure represents the crucial roles played by mitochondria. During disorders, mitochondrial dysfunction can occur due to the alteration of mitochondrial biogenesis, mitochondrial dynamics, ROS production, and mitochondrial-related signaling and apoptosis. ETC, electron transport chain; ACL, ATP citrate lyase; TA, aminotransferase; GLUD1, glutamate dehydrogenase 1; CytC, cytochrome C; TCA, tricarboxylic acid; MOMP, mitochondrial outer membrane permeabilization; ROS, reactive oxygen species; VDAC, voltage-dependent anion channel; SOD, superoxide dismutase; mPTP, mitochondrial permeability transition pore; TRX, thioredoxin; GSH, glutathione; MTHFD, methylene-tetrahydrofolate dehydrogenase; 5,10-CH2-THF, 5,10-methylene-tetrahydrofolate.

**Figure 2 life-12-00657-f002:**
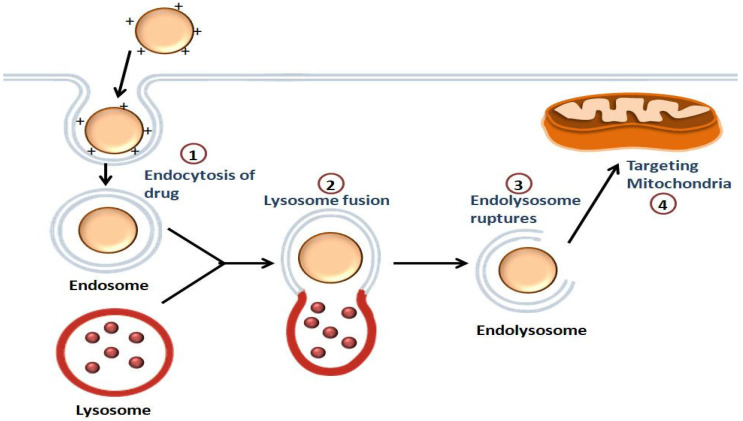
**Illustration of the mitochondrial-targeting of drugs by NPs:** Endocytosis of the drug occurs, followed by the endolysosome formation. The drug is released into the cytoplasm once the endolysosomal membrane is disrupted, and then the drug is targeted to the mitochondria.

**Figure 4 life-12-00657-f004:**
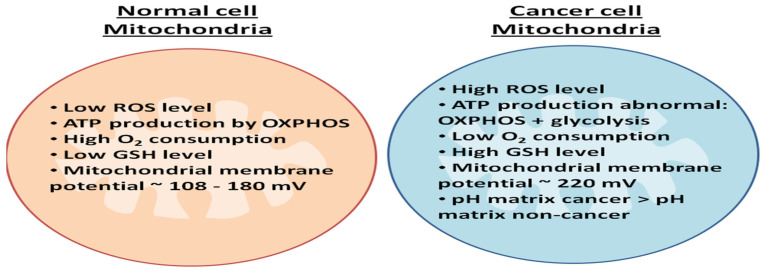
**Difference between normal-cell mitochondria and cancer-cell mitochondria:** Cancer induces changes in normal-cell mitochondria by increasing their ROS level; ATP production occurs by glycolysis; oxygen consumption becomes low; GSH level and membrane potential increase; and the pH of the matrix are greater than the normal cell.

**Figure 5 life-12-00657-f005:**
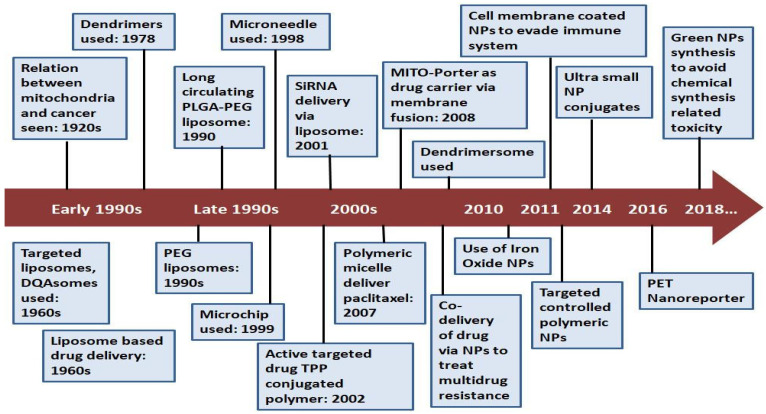
Timeline showing progress in mitochondria-targeting, nano-drug-delivery systems to treat cancer. PEG, polyethylene glycol; PLGA, poly(lactic-co-glycolic acid); TPP, triphenylphosphine; DQA, DeQAlinium; siRNA, small interfering RNA; NP, nanoparticle; PET, positron emission tomography.

**Figure 6 life-12-00657-f006:**
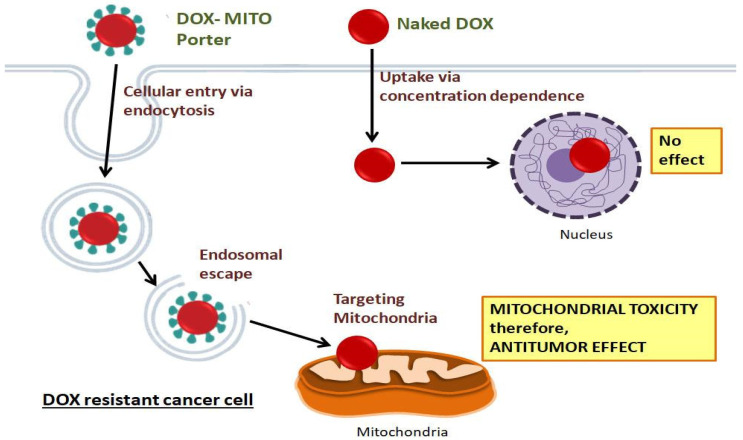
**Use of MITO-Porter to deliver DOX:** Doxorubicin is an anticancer drug. When delivered with a MITO-Porter and targeted to the cell mitochondria, it shows greater antitumor efficacy than naked DOX. The endosomal escape property of the DOX-MITO-Porter showed an antitumor effect via mitochondrial toxicity, while naked DOX when delivered to the cell showed no effect (Adapted from Ref. [90], Yamada et al., 2017).

**Figure 7 life-12-00657-f007:**
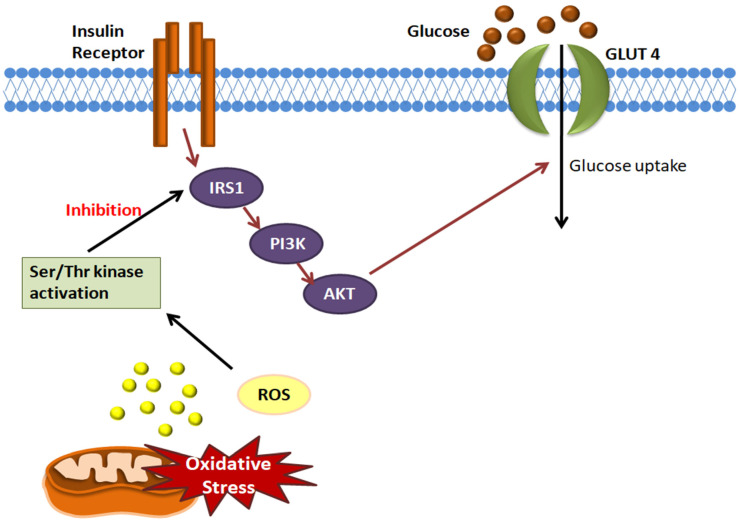
**Association of mitochondrial dysfunction with insulin sensitivity in type 2 diabetes mellitus:** Excessive ROS production occurs due to mitochondrial dysfunction, which leads to the activation of serine/threonine (Ser/Thr) kinases. This leads to the increased serine phosphorylation of insulin receptor substrate (IRS1), which inhibits phosphatidylinositol 3-kinase (PI3K) activity, thus inhibiting glucose uptake.

**Figure 8 life-12-00657-f008:**
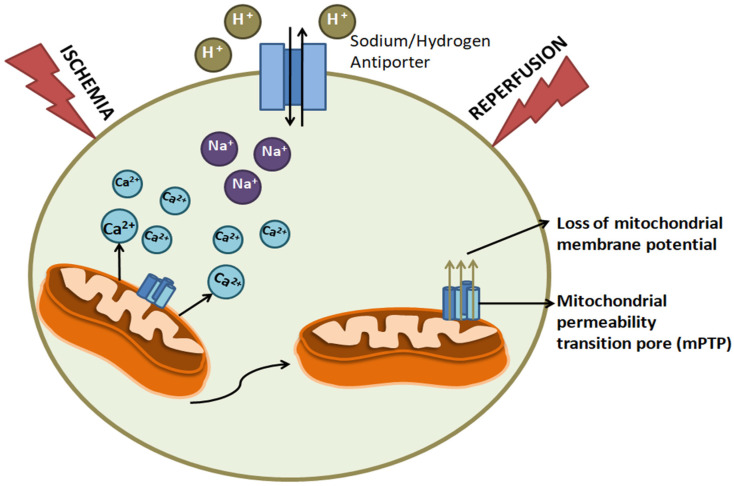
**Process of mitochondrial dysfunction induced by ischemia-reperfusion in cardiomyocytes:** Mitochondria excessively release calcium in the cell. Accumulation of calcium and sodium takes place, which results in the opening of the mPTP, leading to the loss of membrane potential, which is otherwise essential to keep the cells intact.

**Table 1 life-12-00657-t001:** Summary of various mitochondria-targeted drug-delivery systems used for the treatment of cancer, Alzheimer disease, diabetes, and ischemia-reperfusion injury, and their major effects on these pathologies. TPP, triphenylphosphine; AgNP, silver nanoparticle; PAMAM, polyamidoamine; Y_2_O_3_ NPs, yttrium oxide nanoparticles; SLNs, solid lipid nanoparticles; NLCs, nanostructured lipid carriers; WS-CoQ10, water-soluble coenzyme Q10; PLGA, poly(lactic-co-glycolic acid); MCTD-NPs, multistage continuous targeted drug-delivery-carrier nanoparticles; CsA, cyclosporine A;MOMP, mitochondrial outer-membrane permeabilization, Bcl-2: B-cell lymphoma 2; GSH, glutathione; ROS, reactive oxygen species; CytC, cytochrome C; SOD, superoxide dismutase; mPTP, mitochondrial permeability transition pore.

Diseases	Nanosystems	Drug Conjugates	Major Effects	References
Cancer	Liposome	Topotecan	Co-localization in mitochondria, enhanced drug content in mitochondria, dissipated mitochondrial membrane potential, caspase-9- and -3-induced apoptosis	[86]
	TPP-PF127-HA nanomicelle	Paclitaxel	Inhibit antiapoptotic Bcl-2, cause MOMP, caspase-9- and -3-induced apoptosis	[87]
	AgNPs	Extract of plant *Rubus fairholmianus*	ATP and GSH↓, ROS↑, Cyt C release, caspase-3 and -7-activate-induced apoptosis	[88]
	PAMAM dendrimer	Curcumin	ATP and GSH↓, ROS↑, apoptosis, cell cycle arrest at G2/M phase	[89]
	MITO-Porter	Doxorubicin	mitochondrial function↓(mitochondrial membrane potential↓, respiratory activity↓), ATP production↓, cell death	[90]
	Graphene quantum dots	Ruthenium nitrosyl	Photothermal effect	[91]
Alzheimer disease	SLNs	Ferulic acid	Restored mitochondrial membrane potential, ROS↓, Cyt C release↓, mitochondrial membranes stability↑, protective activity on neurons↑	[92]
	NLCs	Curcumin	Oxidative stress↓, Aβ plaque-formation↓	[93]
	Transferrin liposome	Osthole	Oxidative stress↓, lipid oxidation↓, accumulation of Osthole in brain↑, Aβ plaque-deposition↓, neuroinflammation↓	[94]
	WS-CoQ10 nanomicelle	Coenzyme Q10	ROS↓, postponed premature senescence, resumption of autophagy	[95]
	PLGA-b-PEG-TPP	Curcumin	Endosomal and lysosomal escape↑, protective activity on neurons↑	[96]
	PAMAM dendrimer	N-acetyl cysteine	Oxidative stress↓, protective activity on neurons↑	[97]
	Ceria NPs	Ceria	ROS↓, mitochondrial stability↑	[98]
Diabetes	TPP-SLNs	*Ficus religiosa* L.extract	Ca2+ concentration↑, membrane potential↑, antioxidants↑, complex I, II, IV, and V activity↑ apoptotic markers↓, blood glucose↓, glycosylated Hb↓	[99]
	Y_2_O_3_ NPs	-	Oxidative damage↓	[100]
	Lipophilic cation	MitoQ	Restores ATP, ADP, AMP, and cAMP levels;DHAP and glucose-6-phosphate levels↓	[101]
	Pectin NPs	Metformin	Oxidative damage↓, lipid peroxidation↓, glucose absorption↑	[102]
Ischemia-reperfusion injury	MCTD-NPs	Resveratrol	ROS↓, inhibits mPTP opening, inhibits mitochondria-dependent apoptotic pathway, infarct size ↓	[103]
	PLGA CsA-NPs/Pitavastin NPs	Cyclosporine A/Pitavastin	Inhibits mPTP opening and monocyte-mediated inflammation	[104]
	PLGA NP	Quercetin	Oxidative stress↓, cell viability↑	[105]
	PLGA NP	Melatonin	Oxidative stress↓, restored catalase, SOD, GSH activities and lipid peroxidation, mitochondrial membrane stability↑	[106]
	PLGA NP	Mdivi1	Cardioprotection, Inhibition of MOMP and apoptosis	[107]

**Table 2 life-12-00657-t002:** Some metallic NPs used to treat different types of cancer, along with the drug delivered.

Nanoparticle	Drug Delivered	Anti-Cancer Effect against	References
Fe3O4-silica	L-Asparaginase	Acute lymphoblastic leukemia	[116]
SPION-PEG, PEI	Folic acid, doxorubicin	Breast cancer	[117]
AuNP	doxorubicin	Breast cancer	[118]
AuNP	p12	Breast cancer	[119]
AuNP	Paclitaxel	Breast cancer, lung cancer, osteosarcoma	[120]
AuNP	Herceptin	Breast cancer	[121]
AgNP	Fucan A	Kidney cancer	[122]
AgNP	Alisertib	Glioblastoma	[123]
AgNP	Peptide TAT	Malignant melanoma	[124]

**Table 3 life-12-00657-t003:** Different generations of PAMAM dendrimers and the drugs delivered by them to treat cancer.

PAMAM Dendrimer Generation	Drug Delivered against Cancer	References
Generation 4	Gemcitabine	[126]
Generation 4	Doxorubicin	[127]
Generation 4	Methotrexate	[128]
Generation 5	Methotrexate	[129]
Generation 5	Paclitaxel	[130]

## Data Availability

Not applicable.

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
