# Peer review of "Mitochondria-Targeted, Nanoparticle-Based Drug-Delivery Systems: Therapeutics for Mitochondrial Disorders"

_life, 2022, doi:10.3390/life12050657_

Round 1

Reviewer 1 Report

In the present manuscript, entitled “Mitochondria-targeted nanoparticles based drug delivery systems: Therapeutics for mitochondrial disorders », Sakshi Buchke et al. focus on the use of nanoparticles to target the mitochondria. They wrote a review on the existing nanoparticle-based drug-delivery systems like liposomes, DQAsome, MITO-Porters, micelles, polymeric, metal nanocarriers and quantum dots as mitochondria-targeted strategies. They also discuss their potential applications to improve  treatments for mitochondrial disorders.

My major concern is that this kind of reviews already exist. Since 2020 until now there are about ten similar reviews about Mitochondria-targeted drug delivery systems. Most of them mention applications for cancer or neurodegenerative diseases. In the present review authors discuss application in mitochondrial diseases but it is really a pity that only cancers, neurodegenerative diseases, diabetes mellitus and ischemia-reperfusion injury are mentioned as in the existing reviews. Indeed, most applications are focused on anti-cancer drugs whose goal is to kill mitochondria. An effort could be done to treat the primary mitochondrial diseases. To distinguish itself from other rewiews it would have been better to show the potential applications for the so-called mitochondrial diseases linked to a mutation in a nuclear or mitochondrial gene coding for a mitochondrial protein. These diseases are individually rare but collectively numerous.

In addition, it would have been interesting to mention that a same drug can have an opposite effect with different concentrations. For example, curcumin (mentioned in the review for its anti-cancer properties) at high concentration is deleterious for mitochondria while at low concentration it is beneficial.

Moreover, in this review, only anti-oxydants molecules are mentioned to improve mitochondrial functions. Many other molecules known to improve mitochondrial function do not all have an anti-oxidant effect. It would habe been nice to mention it.

This review is far too limited in term of diseases and in term of drugs.

The Part 3. « Mitochondria-targeted nanoparticles-based drug delivery » is well described, documented and organized but in my opinion, the Part 2. « Mitochondrial dysfunction » is useless.

In the same vein, some figures are useless (Figure 2, Figure 5, figure 8 ?, Figure 9 ?).

There are quite numerous mistakes in figures. In figure 1 TCA cycle and ETC should not be outside the mitochondria. The mitochondria knowledge are quite poor and inexact.

« improving MD » does not make sense.

« MD namely, mitochondrial DNA mutations » is not true. There are also muations in nuclears genes encoding mitochondrial proteins.

The last point that is addressed in the conclusion about the drug-free approaches is not clear. It would deserve to be more detailed.

What is very annoying is that the English is very bad. This strongly affects the understanding. A major improvement in English is needed.

Author Response

  1. Effects of curcumin at both high and low concentrations have been added in the cancer part.
  2. More studies on mitochondria targeted drug delivery systems have been incorporated in the diseases part. Ferulic acid, CoQ10 and Osthole in alzheimer’s;  MitoQ, Metformin in diabetes;  Melatonin, Mdivi1 and Quercitin have been added.
  3. Figure 1 has been modified and TCA and ETC are now inside the mitochondria.
  4. Improving MD has been changed to preventing mitochondrial dysfunction.
  5. Both mitochondrial and nuclear DNA mutations have been stated in diabetes.
  6. The last paragraph of conclusion is entirely based on drug-free approaches. Kindly let us know if more details are to be provided in that part.
  7. Several grammatical errors have been corrected.

Reviewer 2 Report

This well-written manuscript is a comprehensive review, which focuses on how nanoparticle based drug delivery systems can be applied to treat mitochondrial dysfunction. The topic of the review is clinically important and the relevant findings of the literature are discussed in a logical order with sufficient numbers of references. However, a few points should be checked before the final publication of the manuscript.

Specific suggestions:

  1. All abbreviations within Figures 1, 6, and 8 should be defined in the corresponding legends.
  2. Charges should be properly shown in Figures 2 and 9.
  3. The subfigures 4B and C should be changed to follow the logic of the text.
  4. The citation found in the legend of Figure 4 (lane 401) should be given with a number.
  5. In Figure 5, “pH matrix, cancer ~ 8.0” should not be inserted to “Normal Cell Mitochondria”.
  6. Figure 6 is not cited in the manuscript text.
  7. In Figure 8, the link between the boxes of “Ser/Thr kinase activation” and “IRS1” should label inhibition.
  8. Lane 420, should not be anaerobic glycolysis? Please, check.
  9. The sentence written in lanes 591-593 should be rephrased.
  10. All abbreviations should be given when first mentioned in the text. Then a defined abbreviation should be consequently used throughout the text.
  11. There are minor grammatical or spelling errors in lanes 1, 37, 39, 110, 120, 202, 214, 234, 241, 659, 660, 677, and 703.

Author Response

  1. Figures 1, 6 (now figure 5), and 8 (now figure7) have been defined in the corresponding legends.
  2. Figure 2 has been removed and charges in figure 9 (figure 8) have been modified.
  3. The text has been modified a little according to the subfigures 4B and C.
  4. Citation in the legend of Figure 4 (now figure 3) has been given with a number.
  5. “pH matrix, cancer ~ 8.0” has been removed from“Normal Cell Mitochondria” in figure 5 (now figure 4)..
  6. Figure 6 (now figure 5) has been added in the manuscript text.
  7. In Figure 8 (now figure 7), the link between the boxes of “Ser/Thr kinase activation” and “IRS1” has been labeled inhibition.
  8. In cancer, it is aerobic glycolysis.
  9. The sentences in the diabetes part have been rephrased.
  10. Changes asked about the abbreviations have been made.
  11. Grammatical and spelling errors in the lines mentioned have been corrected.

Reviewer 3 Report

This is a well-researched review manuscript on the application of mitochondria-targeted nanoparticles based drug delivery system as a therapy for mitochondrial disorders. The authors first described the mitochondrial damage observed in mitochondrial disorders. Then, they described the several instances on how nanoparticles can deliver drugs to the mitochondria. Finally, they described the use of nanoparticles in several pathologies such as cancer, alzheimer’s disease, diabetes and ischemia-reperfusion injury. Overall, the manuscript was written in good English and easy to follow. However, there are still minor grammatical errors and I highly recommend that the revised manuscript to be proofread. Some of my comments that would improve the overall quality of the manuscript are as follow:

  1. The abbreviation “MD” for mitochondrial dysfunction is not very common.
  2. Figure 1 is not mentioned anywhere in the text. 
  3. Please be consistent on how you cite the references. For instance, Line 554, Marrache et al., (2012) or Lines 322-325, Biswas et al…………… [70].
  4. It would be beneficial and helpful to the readers if a summary table describing the applications of nanoparticles in various pathologies could be provided.
  5. In the figures that describe the underlying mechanisms in specific disease, you could potentially list the nanoparticles and their target pathways in the figures.

Author Response

  1. Abbreviation MD for mitochondrial dysfunction has been replaced with mitochondrial dysfunction.
  2. Figure 1 has been mentioned in the introduction part.
  3. Changes asked for citing the references have been done and all the references in the text have been cited in a similar pattern.
  4. A summary table has been added mentioning all the studies stated in the paper.
  5. Since, a few figures in the diseases part were related to mitochondrial dysfunction in the pathologies that’s why we have not added nanoparticles and pathways in them. However, we have stated the effects of the nanoparticles in the summary table.
  6. Several grammatical errors have been corrected.

Round 2

Reviewer 1 Report

The manuscript was improved enough to be published in Life.

Therefore, I suggest the current version of this manuscript for publication in Life.

Reviewer 2 Report

The manuscript was substantially improved. Therefore, I suggest the current version of this manuscript for publication in Life.